# Phytochemical Composition and Antioxidant Activity of Manuka Honey and Ohia Lehua Honey

**DOI:** 10.3390/nu17020276

**Published:** 2025-01-13

**Authors:** Iulia Ioana Morar, Raluca Maria Pop, Erik Peitzner, Floricuța Ranga, Meda Sandra Orăsan, Andra Diana Cecan, Elisabeta Ioana Chera, Teodora Irina Bonci, Lia Oxana Usatiuc, Mădălina Țicolea, Anca Elena But, Florinela Adriana Cătoi, Alina Elena Pârvu, Mircea Constantin Dinu Ghergie

**Affiliations:** 1Pathophysiology, Department of Morpho-Functional Sciences, Faculty of Medicine, University of Medicine and Pharmacy “Iuliu Hațieganu”, 400012 Cluj-Napoca, Romania; iulia.morar@umfcluj.ro (I.I.M.); peitzner.erik@proton.me (E.P.); orasan.meda@umfcluj.ro (M.S.O.); andra.cecan@umfcluj.ro (A.D.C.); chera.elisabeta@umfcluj.ro (E.I.C.); adam.teodora@umfcluj.ro (T.I.B.); lia.usatiuc@umfcluj.ro (L.O.U.); madalina.ticolea@umfcluj.ro (M.Ț.); anca.but@umfcluj.ro (A.E.B.); adriana.catoi@umfcluj.ro (F.A.C.); parvualinaelena@umfcluj.ro (A.E.P.); 2Pharmacology, Toxicology and Clinical Pharmacology, Department of Morpho-Functional Sciences, Faculty of Medicine, “Iuliu Hațieganu” University of Medicine and Pharmacy, 400012 Cluj-Napoca, Romania; 3Food Science and Technology, Department of Food Science, University of Agricultural Science and Veterinary Medicine Cluj-Napoca, Calea Mănăștur, No 3-5, 400372 Cluj-Napoca, Romania; florica.ranga@usamv-cluj.ro; 4Orthodontics, Department of Conservative Odontology, Faculty of Dental Medicine, “Iuliu Hațieganu” University of Medicine and Pharmacy, 400012 Cluj-Napoca, Romania; ghergie.mircea@umfcluj.ro

**Keywords:** Manuka honey, Ohia Lehua honey, inflammation, oxidative stress, antioxidant, polyphenols

## Abstract

Honey is abundant in bioactive compounds, which demonstrate considerable therapeutic effects, particularly on oxidative stress and inflammation. Objectives: This work sought to evaluate the antioxidant mechanisms of Manuka honey (MH) and Ohia Lehua honey (OLH), correlating them with phytochemical analyses in a rat model of experimentally induced inflammation. Methods: The identification of polyphenolic compounds in the extracts was carried out using HPLC-ESI MS. The extracts’ antioxidant activity was evaluated in vitro through DPPH, FRAP, H_2_O_2_, and NO scavenging assays, while in vivo assessments included measurements of total oxidative status (TOS), total antioxidant capacity (TAC), oxidative stress index (OSI), advanced oxidation protein products (AOPP), malondialdehyde (MDA), nitric oxide (NO), and total thiols (SH). Results: The phytochemical analysis found a rich content of phenolic compounds in MH and lower quantities in OLH. In terms of in vitro activity, both MH and OLH exhibited strong DPPH radical scavenging abilities, effective NO and H_2_O_2_ scavenging capacities, and high FRAP-reducing power. In vivo, OLH proved highly effective in enhancing antioxidant capacity and lowering oxidative stress markers, showing significant increases in TAC and substantial reductions in TOS and OSI levels. Conversely, MH displayed limited and dose-dependent antioxidant activity, a considerable increase in TAC and SH, and a moderate decrease in TOS and OSI levels. Conclusions: To our knowledge, this is the first study to assess the phenolic content of OLH and to show its capacity to scavenge free radicals and reduce oxidative stress. The effectiveness of MH primarily relies on its increased antioxidant properties and depends on concentration. These results highlight the importance of investigating natural products in developing antioxidant strategies.

## 1. Introduction

Honey, a natural and ancient remedy, is recognized globally for its diverse nutritional and medicinal properties. Comprising a complex mixture of sugars (80%)—primarily fructose (33–43%), glucose (25–35%), and sucrose (0.2–2%)—along with water (17–20%), polyphenols (2–42 mg GAE/100 g), amino acids (0.2–0.4%), minerals (0.1–0.5%), vitamins, organic acids (0.2–0.8%), enzymes and bioactive compounds, like hydrogen peroxide (H_2_O_2_) and methylglyoxal (MG), honey’s composition varies depending on factors such as botanical source, geographical location, and environmental conditions [1,2,3,4,5]. These bioactive components provide honey with remarkable therapeutic effects, particularly in the context of inflammation and oxidative stress. In recent years, scientific investigations have highlighted honey’s important anti-inflammatory and antioxidant properties with health-promoting effects [6].

Oxidative stress and inflammation are closely linked biological processes, which play pivotal roles in various chronic diseases, including diabetes, hypertension, cancer, and cardiovascular diseases. Oxidative stress occurs when the equilibrium between reactive oxygen species (ROS), which are generated by immune cells such as macrophages and neutrophils during inflammation, disrupts the body’s antioxidant defenses, resulting in cellular damage. This excess of ROS harms proteins, lipids, and DNA and amplifies inflammation by triggering immune responses. Inflammation, on the other hand, is the body’s defense mechanism against harmful stimuli like infections or injuries [7]. While acute inflammation is essential for healing, chronic inflammation can contribute to long-term tissue damage and disease development.

Several molecular pathways are involved in mediating the inflammatory response, including the nuclear factor-kappa B (NF-κB), cyclooxygenase (COX), and mitogen-activated protein kinase (MAPK) pathways. These pathways control the production of pro-inflammatory cytokines, including interleukin-6 (IL-6) and tumor necrosis factor-alpha (TNF-α), which, in turn, amplify the inflammatory response [8]. Meanwhile, NF-κB and oxidative stress can activate the NLRP3 inflammasome, a vital component of the innate immune system, which amplifies inflammation and is implicated in diseases like atherosclerosis and type II diabetes [9].

Studies have demonstrated honey’s ability to influence the production of these key inflammatory markers. For example, thyme honey has been shown to increase the expression of TNF-α and COX-2, while Manuka and Kanuka honey have demonstrated their effectiveness in modulating inflammatory responses in both immune and cancer cell lines [1].

In addition to its anti-inflammatory properties, honey exhibits significant antioxidant activity due to its rich composition of phenolic acids. Some phenolic acids (e.g., ellagic acid, caffeic, p-coumaric, and ferulic acids) and flavonoids (e.g., hespereti, kaempferol, and quercetin), enable it to scavenge free radicals and prevent oxidative damage [6,10].

Lastly, honey is widely recognized for its potent antimicrobial properties, being effective against a range of multidrug-resistant bacteria, including *Pseudomonas aeruginosa* and *Staphylococcus aureus* [10,11]. Its low pH, high sugar content, and bioactive components create a hostile environment for microbial growth. Furthermore, honey’s high viscosity helps protect wounds from infection, promoting healing and tissue repair [10]. Based on their main antimicrobial mechanism, there are two types of honey: peroxide and non-peroxide honey. The antimicrobial mechanism of the peroxide type of honey depends on the glucose oxidase, which, in the presence of water, converts the glucose from honey into gluconic acid and hydrogen peroxide. The antimicrobial mechanism of the non-peroxide type of honey is based on other molecules, like polyphenols, and inhibition of glucose peroxidase by methylglyoxal formation. Manuka honey is the best-known non-peroxide type of honey [12].

The properties of MH were studied beginning in the early 90s, and it continues today, but OLH’s medical properties were not studied [13]. Therefore, the current work aimed to investigate and compare the mechanisms of the antioxidant effects of MH and OLH in correlation with the polyphenol analysis.

## 2. Materials and Methods

### 2.1. Chemicals

Ferrous ammonium sulfate, Ortho-dianisidinedihydrochloride (3-3V-dimethoxybenzidine), Sodium citrate, Reduced glutathione (GSH), Ethylenediaminetetraacetic acid (EDTA), Vitamin C (L(+) ascorbic acid), 5,5V-dithiobis-(2-nitrobenzoic acid) (DTNB), Ribose, Glucose, 2,2V-azino-bis(3-ethylbenz-thiazoline-6-sulfonic acid) (ABTS), Saccharose, (F)-catechin, Potassium persulfate, Hydrogen peroxide (H_2_O_2_), tert-butyl hydroperoxide, Cumenehydroperoxide, Thiobarbituric acid (TBA), 3,5,3′,5′-tetramethylbenzidine (TMB), Xylenol orange [o-cresosulfonphthalein-3,3-bis(sodium methyl iminodiacetate)], Glycerol, Sorbitol, Sulfuric acid, Hydrochloric acid, 1,1,3-3-tetramethoxypropane (malondialdehydebis(Dimethyl Acetal), Alchilamine (N-N-diethyl-para-phenylenediamine, DEPPD), Sodium azide, Butylated hydroxytoluene (BHT), Ortho-dianisidinedihydrochloride (3-3′-dimethoxybenzidine), Ferric chloride, Ferrous ammonium sulfate, Horseradish peroxidase, Sulfanilamide (SULF), N-(1-Naphthyl) ethylenediaminedihydrochloride (NEDD), Sodium nitrite (NaNO_2_), Tris-HCl, EDTA, Methanol, Sodium borohydrate (NaBH4), Sodium chloride (NaCl), Formaldehyde, Chloramine-T, Phosphate-buffered saline (PBS), Distilled water, Potassium iodide (KI). All chemicals were purchased from Sigma Co. (St. Louis, MO, USA) and Merck Co. (Darmstadt, Germany) and were of ultra-pure grade.

The HPLC-grade acetonitrile was acquired from Merck (Darmstadt, Germany), while ultrapure water was obtained with the Direct-Q UV system from Millipore (Burlington, MA, USA). Standard substances, including catechin, luteolin, rutin, chlorogenic acid, gallic acid, hesperidin, and caffeine (all of 99% HPLC grade), were purchased from Sigma (St. Louis, MO, USA).

### 2.2. Honey Products

The Manuka honey (MH) is 100% sourced from honeybees in New Zealand, primarily feeding on the Manuka shrub (*Leptospermum scoparium*; Family: *Myrtaceae*) (Appendix A). This monofloral honey is reportedly “briefly stirred, hand-sifted and bottled” without further processing. The MH used in this study had a Methylglyoxal (MGO) concentration of 400+, as verified by the German manufacturer “HonigWernet”. According to the company, the MGO content is determined in specialized honey laboratories as part of their quality control process.

The Ohia Lehua blossom honey (OLH) (*Metrosiderospolymorpha*; Family: *Myrtaceae*), is sourced directly from the Ka’u District on the Big Island of Hawaii by the “Big Island Bees” honey manufacturer (Appendix A). It is a monofloral honey produced “without chemicals, artificial feeds, miticides, heat, or filtration”, ensuring it is neither heated nor blended.

The fact that both honeys were claimed to be raw and unprocessed is critical, as any processing could have potentially affected the study’s outcomes.

### 2.3. Phytochemical Analysisubsection

#### 2.3.1. Total Polyphenol Content

The total phenolic content (TPC) was evaluated with Folin–Ciocâlteu’s phenolic reagent (Alpha Chemical, Navi Mumbai, India) through a modified protocol of the Folin–Ciocalteu method. The standard curve of gallic acid was used for quantification (Sigma Aldrich, St. Louis, MO, USA). Each sample underwent analysis in quadruple. In summary, 1 mL of honey solution (100 mg/mL diluted in distilled water) from each sample was mixed with 5 mL of 10% Folin–Ciocalteu’s phenol reagent. After a 5 min interval, 4 mL of 7.5% sodium bicarbonate was added, and the mixture was allowed to rest at room temperature for 30 min (in dark conditions). The absorbance was measured at 765 nm with a spectrophotometer. The total phenolic content was calculated using the standard curve of gallic acid solutions (Appendix A) and quantified as milligrams of gallic acid equivalents (GAE) per gram of honey (mg GAE/g) [14,15].

#### 2.3.2. Total Flavonoid Content (TFC)

The total flavonoid content was determined using a previously described method [16]. In brief, 25 μL of honey solution (100 mg/mL diluted in distilled water) was combined with 8 μL of 7% NaNO_2_, 15 μL of 10% AlCl_3_ solution, 50 μL of 1M NaOH solution and 28 μL of distilled water. After thorough mixing, the solution was left at room temperature for 15 min. Absorbance was then measured at 510 nm. Flavonoid content was quantified using calibration curves ranging from 10 to 500 μg/mL, with the results expressed as milligrams of quercetin equivalents per gram of honey (mg QE/100 g) [17].

#### 2.3.3. High-Performance Liquid Chromatography Coupled with Electrospray Ionization Mass Spectrometry (HPLC-ESI MS) Analysis

An Agilent 1200 HPLC system (Agilent Technologies, CA, USA) was used to characterize the honey extracts; as previously described, it was equipped with a quaternary pump, solvent degasser, autosampler, UV-Vis detector with photodiode array (DAD), and a single quadrupole mass spectrometer (MS) detector, model 6110 [18]. The separation of compounds was performed on a Kinetex XB C18 column (4.6 × 150 mm, 5 μm particle size; Phenomenex, USA). Two mobile phases were used. Mobile phase (A), which consisted of water with 0.1% acetic acid, and mobile phase (B), which consisted of acetonitrile with 0.1% acetic acid. Each run lasted 30 min. The temperature was set at 25 °C, and the flow rate at 0.5 mL/min. The elution program started at 0 min with 5% B, continued for 2 min with 5% B; increased up to 18 min from 5% to 40% B; further increased from 18 to 20 min from 40% to 90% B and then kept for 4 min at 90% B. Next, from 24 to 25 min, the percentage was decreased from 90% to 5% B and maintained for another 5 min at 5% B. Spectral data were collected in the 200–600 nm range for all peaks, and chromatograms were recorded at 280 and 340 nm wavelengths.

In the MS analysis, ESI positive full scan ionization mode was used using the capillary voltage at 3000 V, the temperature at 350 °C, the nitrogen flow at 7 L/min, and the mass range between 120 and 1200 *m*/*z*. Data acquisition and interpretation were performed using Agilent ChemStation software, version B.02.01-SR2 [18].

Calibration curves for quantifying phenolic compounds were generated by injecting five different concentrations of each standard dissolved in methanol. The equations derived from these curves were used for the quantitative analysis of each phenolic compound. Hydroxycinnamic acids were quantified as chlorogenic acid equivalents (y = 22.585x − 36.728, (R^2^ = 0.9937), LOD = 0.41 μg/mL, LOQ = 1.24 μg/mL)); hydroxybenzoic acids as gallic acid equivalents(y = 33.624x + 30.8; R^2^ = 0.9978; LOD = 0.35 µg/mL, LOQ = 1.05 µg/mL); flavones as luteolin equivalents (y = 68.857x + 25.113, (R^2^ = 0.9972), LOD = 0.38 μg/mL, LOQ = 1.14 μg/mL)); flavanones as hesperidin equivalents (y = 11.206x + 77.19, R2 = 0.9968; LOQ = 0.85, LOD = 2.55 µg/mL), and flavonols as rutin equivalents (y = 26.935x − 33.784, (R2 = 0.9981), LOD = 0.21 μg/mL, LOQ = 0.64 μg/mL)).

The identification of phenolic compounds was performed by comparing their mass spectra, retention periods, and UV-Vis absorption with reference standards, published data, and information from the Phenol-Explorer database.

### 2.4. In Vitro Antioxidant Activity Analysis

#### 2.4.1. DPPH Radical-Scavenging Activity

The DPPH radical-scavenging activity of the honey was assessed using a previous method [19]. A 1,1-diphenyl-2-picrylhydrazyl (DPPH) assay was employed for the analysis. This procedure combined 100 μL of the honey solution with 100 μL of the DPPH working solution. The mixture was then incubated in the dark for 30 min, and the absorbance was measured at 517 nm to determine the percentage of DPPH radical-scavenging activity. The DPPH scavenging activity (AA) was computed as a percentage using the formula AA% = [(A control − A sample)/A control] × 100, where A control is the absorbance of DPPH radical + methanol and A sample is the absorbance of DPPH radical+ sample extract. The IC50 value, indicating the concentration required to inhibit 50% of DPPH free radicals, was converted to μgTrolox equivalents/mL (μg TE/g).

#### 2.4.2. Ferric Ion Reducing Antioxidant Power Assay (FRAP)

The Ferric Ion Reducing Antioxidant Power (FRAP) assay used a previous procedure [20]. Moreover, 10 μL of each honey solution was combined with 190 μL of the FRAP reagent. After incubating for 30 min in the dark, the absorbance was measured at 593 nm using a microtiter plate reader. FRAP was calculated with the formula: (C_TX_x V × D × 100)/weight of the honey sample, where C_TX_ is the TX concentration (mg/mL) from the standard TX curve, V (mL) is the honey volume, and DF is the dilution factor. Results were expressed as μg TE/g.

#### 2.4.3. Hydrogen Peroxide (H_2_O_2_) Scavenging Activity

The ability of honey solutions to scavenge hydrogen peroxide (H_2_O_2_) was evaluated following a previously described method [21]. In short, the extracts were combined with an H_2_O_2_ solution, and the absorbance was measured at 230 nm against a phosphate buffer blank after 10 min. The IC50 of H_2_O_2_ scavenging was calculated using the formula: scavenged H_2_O_2_ % = [(A control − A sample)/A control] × 100, where A control is a solution containing phosphate buffer and hydrogen peroxide, and A sample is a solution containing phosphate buffer, hydrogen peroxide, and serum sample. The results were converted to mg TE/g.

#### 2.4.4. Nitric Oxide (NO) Radical Scavenging Assay

The nitric oxide radical scavenging assay was conducted as previously described [21]. Using sodium nitroprusside to generate nitric oxide (NO), the Griess reagent was used to detect nitric oxide (NO). Briefly, 0.5 mL of honey solutions were mixed with 2 mL sodium nitroprusside solution (SNP) and 0.5 mL of PBS (pH 7.4). The mixture was then incubated at 25 °C for 2.5 h. Next, 0.5 mL of the mixture was mixed with 1 mL of sulphanilic acid. After 5 min, 1 mL of Naphthylethylenediaminedihydrochloride was added. Afterward, the mixture was vortexed and incubated in dark conditions for 30 min. Absorbance was measured at 546 nm, and IC50 was calculated using the formula: scavenged NO % = [(A blank − A sample)/A blank] × 100. The IC50 results were expressed as micrograms of quercetin equivalent per g (µg QE/g).

The in vitro antioxidant analysis was performed in triplicate. The measurements were performed with a UV-Vis spectrophotometer (Jasco V-350, Jasco International Co., Ltd., Tokyo, Japan).

### 2.5. In Vivo Experimental Design

#### 2.5.1. Animal Subjects

This experiment was conducted on male albino Wistar rats weighing 200–250 g. The animals were obtained from the Animal Facility of “Iuliu Haţieganu“ University of Medicine and Pharmacy, Cluj-Napoca, Romania. They were accommodated in polypropylene cages and held under controlled environmental conditions: 12:12 h light:dark cycle, temperature 25 ± 10 °C and relative humidity 55 ± 5%. *Ad libitum* water and a normal granular diet were freely available.

All procedures adhere to Directive 2010/63/EU and Romanian national law 43/2014 regarding protecting animals used in scientific research. The project received approval from the Veterinary Sanitary Direction and Food Safety Cluj-Napoca (Approval No. 372/04.07.2023). The experiments were performed in triplicate.

#### 2.5.2. Experimental Protocol

The animals were randomized and put into 10 groups (*n* = 5): CONTROL group, inflammation group (INFL), group INFL treated with diclofenac (DICLO) (10 mg/kg) [22], group INFL treated with TROLOX (10 mg/kg) [23], 3 groups INFL treated with 1 mL/rat/day of three MH (*Leptospermum scoparium*) dilutions, respectively 100% (MH100) (2 g honey/kg b.w./day), 50% (MH50) (1 g honey/kg b.w./day), and 25% (MH25) (0.5 g honey/kg b.w./day), and 3 groups INFL treated with three OLH (*Metrosiderospolymorpha*) dilutions, respectively 100% (OLH100) (2 g honey/kg b.w./day), 50% (OLH50) (1 g honey/kg b.w./day) and 25% (OLH25) (0.5 g honey/kg b.w./day). Except for CONTROL animals, all INFL groups received intramuscular turpentine oil (6 mL/kg b.w.) on the first day. Subsequently, from the second day onwards, the animals were administered treatments orally via gavage for 10 days. One hour before the administration, honey was prepared daily by solving each dose/rat/day in 1 mL of distilled water [24]. CONTROL and INFL groups received tap water (1 mL/rat/day). On the 12th day, the rats were sedated with ketamine (60 mg/kg) and xylazine (15 mg/kg) [25]. Blood was collected via retro-orbital sinus puncture. Serum was immediately separated and stored at −80 °C until further analysis. The animals were then humanely euthanized by cervical dislocation under general anesthesia, following ethical guidelines approved by the Ethics Committee.

#### 2.5.3. Assessments of Oxidative Stress Markers

Determination of Total Antioxidant Capacity

Total Antioxidant Capacity (TAC) assesses the ability of an organism’s antioxidants to neutralize harmful free radicals and combat oxidative stress. It was measured using a colorimetric method. A standard solution of Fe^2+^-o-dianisidyl underwent the Fenton reaction with a standard H_2_O_2_ solution, forming hydroxyl radicals, which in the presence of an acid oxidized o-dianisidine to dianisidyl radicals. The antioxidants from the sample inhibited the oxidation reactions and the appearance of coloration proportional to their concentrations [26]. The absorbance was read at 440 nm, and results were reported in millimoles of Trolox equivalents per liter (mmol TE/L) (Appendix A).

Determination of Total Oxidative Status

Total oxidative status (TOS)was measured as a general marker of serum oxidants. It was measured using a colorimetric assay based on the oxidation of ferrous ion (Fe^2+^) to ferric ion (Fe^3+^) in the presence of oxidant species in an acidic medium, where the measurement of the ferric ion was conducted through a reaction with xylenol orange. The absorbance was read at 560 nm, and results were expressed in micromoles of hydrogen peroxide equivalents per liter (µmol H_2_O_2_/L) (Appendix A) [27].

Determination of Oxidative Stress Index

The Oxidative Stress Index (OSI) is a marker used to assess the level of oxidative stress by evaluating the ratio between TOS and TAC, which represent the overall pro-oxidant load and the body’s antioxidant defenses, respectively [28]. This index provides insight into the balance between oxidative damage and the body’s ability to neutralize it. OSI is calculated using the formula: OSI = TOS (μmol H_2_O_2_ Eq/L)/TAC (μmol TE/L).

Determination of Nitric Oxide

Nitric oxide (NO), the naturally occurring signaling molecule produced by nerve, endothelial, and immune cells, was quantified. The quantification was indirectly performed by measuring total nitrites and nitrates with Griess reaction. Accordingly, serum proteins were removed following the methanol/diethyl ether solution extraction (3:1 (*v*/*v*)). The nitrates were converted to nitrites following vanadium (III) chloride addition. After Griess reagent was added, absorbance was read at 540 nm. Results were expressed in micromoles of nitrite per liter (µmol/L) [16,29].

Determination of Malondialdehyde

Malondialdehyde (MDA), considered an important marker of lipid peroxidation, was measured. The thiobarbituric acid method was performed as previously described. Serum (0.1 mL) was combined with 40% trichloroacetic acid (0.1 mL) and mixed with 0.67% thiobarbituric acid (0.2 mL). The mixture was heated for 30 min using a boiling water bath and, afterward, cooled in an ice water bath. Next, the mixture was centrifuged (3461× *g*, 5 min). The absorbance was read at 532 nm, and the MDA concentration in the serum was expressed in nanomoles per milliliter (nmol/mL) [30].

Determination of Advanced Oxidation Protein Products

Advanced oxidation protein products (AOPP), considered biomarkers for protein oxidation, were determined spectrophotometrically [31]. The samples and a blank of chloramine T were diluted (to 10%)in phosphate-buffered saline (PBS). Next, potassium iodide and glacial acetic acid were added. The absorbance of the samples was registered at 340 nm after the addition of glacial acetic acid. Before data analysis, the optical density of the blank sample was subtracted. AOPP concentrations were calculated and expressed in μmol chloramine-T equivalent/L (μmol chloramine E/L).

Determination of Total Thiols

Total thiols (SH), considered a class of sulfur-containing compounds with antioxidant effect, were determined to evaluate the associated antioxidant capacity. Ellman’s reagent (Sigma-Aldrich, Munich, Germany) was used for the assay, and the absorbance of the supernatant was recorded at 412 nm [32]. The concentration of serum SH was expressed as millimole of glutathione per milliliter (mmol GSH/mL).

Spectrophotometric analyses, including TAC, TOS, NO, MDA, SH, OSI, and AOPP, were conducted using a UV-Vis spectrophotometer (Jasco V-350, Jasco International Co., Ltd., Tokyo, Japan). 

### 2.6. Statistical Analysis

The results were presented as the mean ± standard deviation (SD) for data following a normal distribution. Group comparisons were conducted using one-way analysis of variance (ANOVA) followed by post hoc Bonferroni–Holm tests. Correlation analysis was performed using the Pearson test and principal component analysis (PCA). A *p*-value of less than 0.05 was considered statistically significant.

## 3. Results

### 3.1. Phytochemical Analysis

#### 3.1.1. Total Polyphenols and Flavonoid Content

TPC and TFC of both OLH and MH were substantial, with MH exhibiting higher values than OLH (Table 1).

#### 3.1.2. HPLC-ESI-MS Analysis of Phenolic Compounds

MH and OLH provided a different pattern of HPLC-DAD-ESI + results, with MH having about ten times more phenolic compounds. Phenolics analysis showed that honey solutions were rich in non-flavonoid compounds, like hydroxybenzoic and hydroxycinnamic acids (Figure 1; Table 2). From the phenolic acids, some hydroxybenzoic acids, respectively methyl-syringic acid and trimethoxybenzoic acid, were the most abundant in MH and were missing in OLH. From the hydroxycinnamic acids, p-coumaroyquinic acid, chlorogenic acid, and caffeic acid-glucoside were identified only in MH. Other phenolic acids extracted from both MH and OLH include 2,4-dihydroxybenzoic acid, gallic acid, protocatechuic acid, 4-hydroxybenzoic acid, vanillic acid, and syringic acid. From the flavonoids apigenin-glucoside, quercetin-glucoside, and quercetin were found just in MH, pinocembrin-glucoside was found in MH and OLH. Furthermore, MH and OLH had almost similar phenyllactic acid content.

### 3.2. In Vitro Antioxidant Activity

The MH and OLH displayed in vitro antioxidant activity. Regarding antioxidant activities, the concentration of honey necessary to inhibit 50% of DPPH radical was above 100 μg TE/g and smaller than that of Trolox. H_2_O_2_ and FRAP scavenging capacities were more robust than those of Trolox. Compared to quercitin, MH and OLH NO scavenging activity was also higher (Table 3). There were no significant differences between the antioxidant test results of MH and OLH.

### 3.3. In Vivo Antioxidant Activity

Oxidative stress (OS) was evaluated using both global and specific biomarkers. The INFL group exhibited higher OS levels compared to the CONTROL group, marked by a significant increase in the levels of the global OS markers, TOS and OSI, and NO, MDA, and AOPP (*p* < 0.001). Moreover, there was a moderate reduction in TAC (*p* < 0.05) and an important decrease in the SH levels (*p* < 0.01) (Table 4).

Diclofenac administration produced a slight decrease in TOS and OSI (*p* < 0.05), associated with a slight AOPP reduction (*p* < 0.05) and a small SH increase (*p* < 0.05). TROLOX antioxidant activity caused a moderate decrease in TOS, OSI, and AOPP (*p* < 0.01), along with an important reduction of NO (*p* < 0.001), a weak increase in TAC (*p* < 0.05) and a moderate increase in SH (*p* < 0.01) (Table 4).

The administration of OLH significantly increased TAC (*p* < 0.01), and the effects were better than those of diclofenac, with OLH25 having the best activity. OLH reduced TOS and OSI levels (*p* < 0.001), and OLH100 and OLH50 were stronger inhibitors than diclofenac (*p* < 0.01). OLH did not significantly influence NO and AOPP. All three OLH dilutions lowered MDA (*p* < 0.01), and the effect was stronger than those of diclofenac and trolox (*p* < 0.01). SH was increased by OLH treatments (*p* < 0.01), and there were no significant differences when compared to diclofenac and trolox (*p* > 0.05) (Table 4).

None of the three MH dilutions had a notable effect on TAC compared to the INFL group (*p* > 0.05). Regarding TOS, only MH25 caused a moderate reduction of TOS and OSI (*p* < 0.01). Compared to DICLO and TROLOX, all MH100 and MH50 dilutions had lower inhibitory activity on TOS. NO levels were lowered by MH (*p* < 0.01), and the effect was better than that of diclofenac (*p* < 0.05). MH dilutions had no important activity on MDA (*p* > 0.05) but reduced AOPP (*p* < 0.01). The MH50 caused a significant rise in SH levels (*p* < 0.001), and the effect was better than that of diclofenac (*p* < 0.001) (Table 4).

### 3.4. Principal Component Analysis

The PCA analysis was conducted to evaluate the correlations among the analyzed parameters and to assess their variability across rat groups based on different concentrations of honey administration (Figure 2). The variability was evaluated by comparing the first principal components (PC1 and PC2), as seen in the score plots (Figure 2). For OLH administration at concentrations of 100%, 50%, and 25%, these components (PC1 and PC2) accounted for 82.92%, 97.82%, and 87.29% of the total variance, respectively (Figure 2A–C). Similarly, for MH administration at 100%, 50%, and 25% concentrations, PC1 and PC2 explained 80.68%, 88.54%, and 100% of the total variance, respectively (Figure 2D–F).

For OLH100, the PCA analysis indicated a positive correlation between TOS, OSI, and SH. In OLH50, these were also correlated with TAC; in OHL25, they were associated with TAC and MDA (Figure 2).

For MH100, there was a positive correlation between TOS, OSI, SH, and AOPP, and only NO was negatively correlated. In MH50PCA analysis, NO, AOPP, and SH were positively correlated, but TOS and OSI were negatively correlated to NO, AOPP, and SH. PCA analysis in MH25 indicated that only AOPP was negatively correlated, with the rest of the parameters being positively correlated.

These findings indicate that antioxidant and oxidative stress parameters are influenced differently depending on the type and concentration of honey.

## 4. Discussion

Apitherapy is an alternative therapy that uses bee products, such as honey, propolis, royal jelly, pollen, and bee venom [31]. Today, honey is a functional food produced by honeybees (*Apismellifera*) by mixing plant nectar with bee hypopharyngeal excretions. Consuming functional foods provides the body with health benefits such as the reduction of inflammation and oxidative stress and the prevention of neurodegenerative diseases and cancer [3]. The study of honey composition is needed to find the compounds responsible for some of the health-promoting effects [33], like their antibacterial, anti-inflammatory, antioxidant, antithrombotic, antiallergic, antimutagenic, anti-cytostatic and immunosuppressive effects [34]. It contains about 200 substances [31], and the percentages of these differ between different types of honey due to the floral sources, nectars, seasons [35], climates, environmental conditions, genetic factors, and others [36]. Several studies have revealed that the antioxidant capacity of honey correlates with the presence of specific proteins, amino acids, carotenoids, phenolic compounds and flavonoids, ascorbic acid, and organic acids [31,36]. Based on the literature data, vitamins have no significant contribution to honey’s antioxidant capacity [37].

Polyphenols are plants’ secondary metabolites that are transferred to honey, and phenolic acids and flavonoids were the main groups detected in honey [33]. In MH and OLH samples, TPC and TFC were higher than in other honey samples [38]. In this study, the polyphenols of MH and OLH samples were determined by HPLC-DAD-ESI+. Like in other studies [24], the identified polyphenols were phenolic acids, like benzoic and cinnamic acids, and flavonoids, like flavonols, flavones, and flavanones.

The chemical and structural differences in the phytochemical profiles of MH and OLH can strongly influence their bioactivities. MH, with its higher phenolic content and greater diversity of compounds, is likely to exhibit superior antioxidant properties compared to OLH.

MH is characterized by high percentages of hydroxybenzoic acids (89.61% of total phenolics), predominating compounds such as methyl-syringic acid (144.37 μg/mL), syringic acid (80.58 μg/mL), and trimethoxybenzoic acid (106.30 μg/mL). These compounds contribute to strong antioxidant and anti-inflammatory properties. Next, hydroxycinnamic acids were 5.44% of total phenolics with p-coumaroyquinic acid (10.22 μg/mL) and chlorogenic acid (4.50 μg/mL), known for their antioxidant and enzyme-inhibiting activities. Furthermore, flavonoids were present in a smaller percentage (1.80% of total phenolics) with quercetin-glucoside (1.92 μg/mL) and apigenin-glucoside (0.72 μg/mL), which are known as well for their anti-inflammatory and radical-scavenging properties [39,40].

In OLH (45.49 μg/mL), high percentages of hydroxybenzoic acids (67.98% of total phenolics) were also identified (e.g., 2,4-dihydroxybenzoic acid: 7.35 μg/mL, gallic acid: 17.49 μg/mL). Even though OLH lacks hydroxycinnamic acids and flavonoids, other phenolics like phenyllactic acid represent 31.27% of total phenolics, indicating its potential for antioxidant activity [39,40].

The present study highlights that MH mainly contributed to the in vivo antioxidative properties by causing an important increase in SH, and by a smaller reduction of NO and AOPP. OLH in vivo antioxidant activity had another mechanism other than MH; it consisted of an important reduction of TOS, OSI, and MDA and a smaller increase in TAC and SH. It has been previously demonstrated that phenolic compounds may decrease MDA, an important genotoxic product of lipid peroxidation [41]. Also, among the phenolic acids, methyl-syringic acid, trimethoxybenzoic acid, and syringic acid had the highest concentration in MH. Methyl-syringic acid was found in large quantities only in MH. It is the ester of syringic acid, and it has been reported to be one of the abundant constituents of honey. Methyl-syringic acid shows antioxidant and anti-radical activities [42]. Trimethoxybenzoic acid is a potent antioxidant and inhibitor of cytokine production. Syringic acid was found in both honey samples, MH and OLH, but at a higher concentration in MH. It possesses medicinal properties such as antioxidant, antimicrobial, anti-inflammatory, hepatoprotective, cardioprotective, neuroprotective, and anti-diabetic activities [43]. Syringic acid antioxidant and anti-inflammatory activities rely on their ability to neutralize free radicals and to inhibit the NF-κB-iNOS-COX-2 and JAK-STAT signaling pathways [44]. Other phenolic acids were also reported to be found in other honey samples, such as cinnamic acid, coumaric acid, benzoic acid, and chlorogenic acid [35].

The presence of certain flavonoids in honey is also an indicator of its antioxidant activity. The most important flavonoids found in different honey samples are quercetin, myricetin, chrysin, apigenin, luteolin, pinocembrin, pinobanksin, triacetin, kaempferol, naringenin, hesperidin, fisetin, wogonin, genkwanin, acacetin, catechin and epicatechin [35,39]. In our study, MH TFC was higher than that of OLH, apigenin-glucoside, quercetin-glucoside and quercetin were found only in MH and pinocembrin-glucoside was found in MH and OLH [45].

Considering these differences between the antioxidant mechanisms and the phytochemical analysis, further detailed research is required in order to elucidate which compounds have the most significant effect on these oxidative stress biomarkers.

Honey has been used for medical purposes since ancient times, primarily due to its wound healing and antibacterial effects [46]. There is a medical-grade honey (Reva Mil, Medi honey) that has a broad-spectrum bactericidal activity, and it is used as a topical antibacterial prophylaxis or treatment [47]. Both wounds and infections associate an inflammatory response with inflammation-dependent oxidative stress. Therefore, the present study evaluated some of the MH and OLH antioxidant activity mechanisms.

The antioxidant activity of phenolic compounds consists of their ability to be hydrogen donors, reducing agents, metal chelators, and free radical scavengers [44,48,49]. DPPH, FRAP, NO, and H_2_O_2_ scavenging tests are commonly used tests evaluating different antioxidant mechanisms [39]. Regarding MH and OLH, the samples displayed strong DPPH radical scavenging capacity, NO and H_2_O_2_ scavenging capacities, and high FRAP-reducing power activity as compared to those reported by other investigations [34]. These results can be attributed to many factors, including floral origin, harvest season, and environmental conditions. Because in vitro interactions between the bioactive compounds may induce changes in antioxidant capacities [34], we continued by testing the in vivo antioxidant effects in an experimental rat inflammation model.

Although it is known that honey exhibits in vitro antioxidant activities, there is limited information about the specific mechanisms of its in vivo antioxidant capacity [40]. Furthermore, because the main therapeutic effects of honey are attributed to the polyphenols, and polyphenols’ interactions with other food compounds during digestion may influence their bioaccessibility and bioavailability [50], in vivo study is highly needed.

Thus, in this study, we assessed the in vivo antioxidant effect of OLH and MH against oxidative stress markers in an experimentally induced acute inflammation model. The results highlighted the impact of these natural products on oxidative stress. Previous animal studies have illustrated the connection between honey and conditions caused by oxidative stress. The intake of honey appears to boost the activity of important antioxidant enzymes, such as superoxide dismutase (SOD), glutathione-disulfide reductase (GR), catalase (CAT), and glutathione peroxidase (GPx), in the livers of rats that consumed acetaminophen for 10 days [51]. Although the antioxidant capacity of honey is widely acknowledged, the exact mechanisms through which it provides these effects are still not fully understood. Among the most studied interrelated mechanisms are hydrogen donation, metal ion chelation, and the scavenging of free radicals by boosting the levels of essential antioxidant molecules and enzymes, including β-carotene, vitamin C, glutathione reductase, and uric acid [52].

The INFL group demonstrated significantly elevated OS levels, as evidenced by increased TOS, OSI, and markers such as NO, MDA, and AOPP. Similar findings in prior studies have underscored that inflammation exacerbates oxidative stress, leading to cellular damage and contributing to pathological conditions such as arthritis and cardiovascular diseases [53,54]. The observed moderate reduction in TAC and a significant decrease in SH levels in the INFL group suggests that an inflammatory environment can deplete the body’s antioxidant defenses, corroborating evidence from recent literature [55].

Diclofenac, a commonly used non-steroidal anti-inflammatory drug (NSAID), exhibited a modest ability to mitigate oxidative stress, leading to slight decreases in TOS and OSI. This aligns with previous reports indicating that, while NSAIDs can reduce inflammatory responses, their antioxidant activity is often limited [56]. The minor increases in SH levels may illustrate an aspect of diclofenac’s protective mechanisms, yet the effects were insufficient to restore antioxidant balance significantly.

On the other hand, the synthetic antioxidant TROLOX manifested a more pronounced reduction in oxidative stress biomarkers. This finding supports the established role of vitamin E analogs in effectively scavenging reactive oxygen species, thereby protecting cellular integrity and function [57]. The improvement in TAC and SH levels following TROLOX treatment further emphasizes its capacity to enhance antioxidant defenses in vivo.

OLH showed remarkable efficacy in enhancing antioxidant capacity and reducing oxidative stress markers. With significant increases in TAC and substantial decreases in TOS and OSI, especially with OLH25, the data strongly suggest that OLH possesses potent bioactive compounds contributing to its antioxidant properties. Studies have shown that honey’s high phenolic content contributes significantly to its ability to scavenge free radicals and mitigate oxidative damage [52,58]. To our knowledge, this is the first study to evaluate this aspect in OLH. The effectiveness of OLH in reducing MDA levels, an indicator of lipid peroxidation, is significant; it demonstrates that OLH can aid in preserving cellular membrane integrity during inflammatory states, aligning with literature findings that highlight honey’s protective effects against lipid peroxidation [27]. The lack of significant impact on NO levels and AOPP could indicate that OLH’s mechanisms might be more aligned with lipids protection and enhancement of natural antioxidant pathways instead of directly affecting nitric oxide production and protein oxidation.

Additionally, the PCA analysis revealed a significant positive correlation among TOS, OSI, and SH levels in the OLH treatments, indicating that OLH not only diminishes oxidative stress markers but also enhances the antioxidant defense mechanisms. These findings align with the notion that honey may have intricate effects on oxidative pathways, boosting endogenous antioxidant levels while reducing oxidative damage [59].

In contrast, MH showed limited antioxidant activity, as only the MH25 dilution resulted in a moderate decrease in TOS and OSI levels. While MH is well-known for its unique antibacterial properties [60,61], its comparative antioxidant efficacy appears less robust than that of OLH. Previous research indicates variability in the antioxidant properties of different honey types, suggesting that the phytochemical composition and concentration of active compounds in MH may not be uniform, thereby influencing its antioxidant efficacy [61,62]. The notable increase in SH levels observed in MH50 suggests potential protective mechanisms against oxidative stress, yet the overall performance in enhancing TAC was not compelling.

Furthermore, while MH managed to lower NO levels and reduce AOPP, underlying evidence suggests that stresses from inflammatory responses may require more robust antioxidant support than what MH can provide on its own [63]. The correlations noted in the PCA analysis for MH reflect the complex interplay between oxidative markers, indicating that different honey types may interact uniquely with oxidative pathways.

The main limitations of the study were related to the small experimental groups and the phytochemical analysis limited to polyphenol determination.

## 5. Conclusions

In summary, this study is, to our knowledge, the first to provide compelling evidence for the antioxidant potential of OLH, which reduces serum oxidant concentration, positioning it as a promising natural therapeutic agent for managing inflammation-induced oxidative stress. Conversely, while MH offers some benefits, its effectiveness as an antioxidant is based on the phenolic compound content and serum antioxidant increase. However, it may vary based on composition and concentration, suggesting that further research is warranted. These findings underscore the importance of exploring natural products in developing antioxidant strategies, emphasizing their potential alongside established pharmaceutical interventions in treating inflammatory conditions. Future studies should aim to elucidate the specific molecular mechanisms underlying the protective effects of OLH and the variable responses observed with MH, thus enhancing our understanding of their respective roles in oxidative stress reduction.

## Figures and Tables

**Figure 1 nutrients-17-00276-f001:**
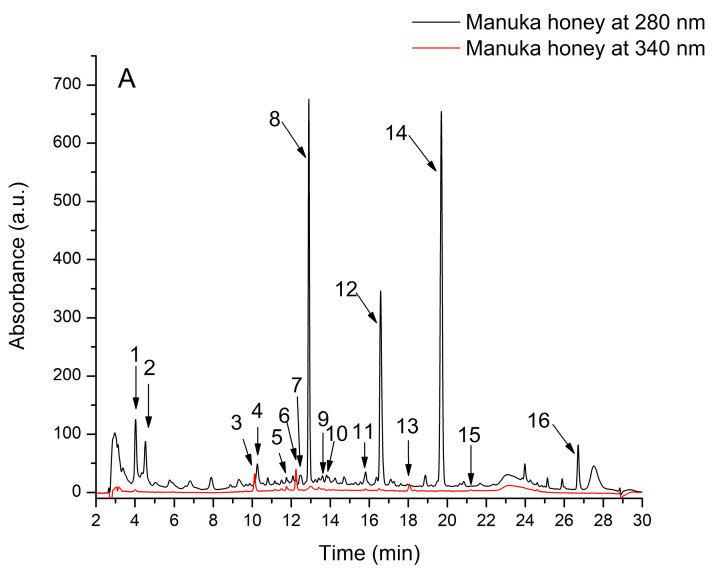
HPLC chromatogram of phenolic compounds from Manuka honey and Ohia Lehua honey solutions (100 mg/mL): (**A**) Manuka honey phenolic compounds at 280 and 340 nm; (**B**) Ohia Lehua honey phenolic compounds at 280 and 340 nm. The peak identification is provided in Table 2.

**Figure 2 nutrients-17-00276-f002:**
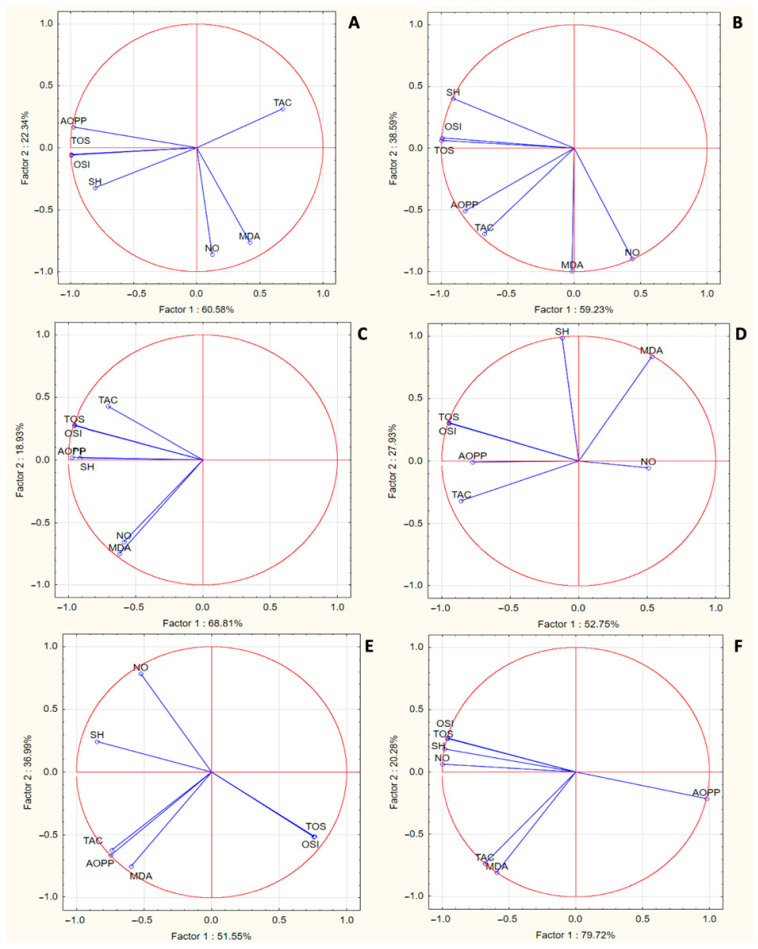
The PCA results of oxidative stress biomarkers based on the correlation matrix with PC1 and PC2 for Ohia Lehua honey (*Metrosiderospolymorpha*) and Manuka honey (*Leptospermum scoparium*): (**A**) PCA of OLH100—Ohia Lehua honey100%; (**B**) PCA of OLH50—Ohia Lehua honey 50%; (**C**) PCA of OLH25—Ohia Lehua honey 25%; (**D**) PCA of MH100—Manuka honey 100%; (**E**) PCA of MH50—Manuka honey 50%; (**F**) PCA of MH25—Manuka honey 25%.

**Table 1 nutrients-17-00276-t001:** Total polyphenols and total flavonoid content of Manuka honey and Ohia Lehua honey.

Plantextract (100 mg/mL)	Total Polyphenols Content (mgGAE/g)	Total Flavonoids Content (mg QE/g)
Ohia Lehua honey	4.826 ± 00.4	12.65 ± 1.05
Manuka honey	5.425 ± 0.09	31.65 ± 1.86

Note: Values are expressed as mean ± SD (n = 3).

**Table 2 nutrients-17-00276-t002:** HPLC-DAD-ESI + phenolic compound tentative identification from Manuka honey and Ohia Lehua honey solutions (100 mg/mL).

Peak No	R_t_ (min)	UV λ_max_ (nm)	[M + H]^+^ (*m*/*z*)	Compound	Subclass	Manuka Honey (μg/mL)	Ohia Lehua Honey (μg/mL)
1	4.03	270	155	2,4-Dihydroxybenzoic acid	Hydroxybenzoic acid	36.65 ± 2.12	7.35 ± 0.93
2	4.53	270	171	Gallic acid	Hydroxybenzoic acid	30.04 ± 2.35	17.49 ± 1.46
3	10.12	320	339	p-Coumaroyquinic acid	Hydroxycinnamic acid	10.22 ± 0.04	ND
4	10.25	280	155	Protocatechuic acid	Hydroxybenzoic acid	18.98 ± 4.71	4.71 ± 0.05
5	11.76	330	355	Chlorogenic acid	Hydroxycinnamic acid	4.50 ± 0.02	ND
6	12.24	330	343	Caffeic acid-glucoside	Hydroxycinnamic acid	11.54 ± 1.30	ND
7	12.49	270	139	4-Hydroxybenzoic acid	Hydroxybenzoic acid	9.94 ± 0.97	0.72 ± 0.22
8	12.90	265	213	Trimethoxybenzoic acid	Hydroxybenzoic acid	106.30 ± 4.20	ND
9	13.42	340, 245	433, 271	Apigenin-glucoside	Flavone	0.72 ± 0.04	ND
10	13.81	280	169	Vanillic acid	Hydroxybenzoic acid	6.14 ± 3.02	0.37 ± 0.01
11	15.81	360, 250	465, 303	Quercetin-glucoside	Flavonol	1.92 ± 0.90	ND
12	16.58	280	199	Syringic acid	Hydroxybenzoic acid	80.58 ± 5.68	0.28 ± 0.01
13	18.05	350, 250	419, 257	Pinocembrin-glucoside	Flavanone	4.30 ± 2.89	0.34 ± 0.05
14	19.69	280	213, 199	Methyl-Syringic acid	Hydroxybenzoic acid	144.37 ± 6.21	ND
15	21.21	360, 250	303	Quercetin	Flavonol	1.74 ± 0.98	ND
16	26.71	280	167	Phenyllactic acid		14.35 ± 2.69	14.23 ± 2.24
				Total phenolics		482.30 ± 34.43	45.49 ± 4.95

Note: Values are expressed as mean ± SD (n = 3); ND—not detected.

**Table 3 nutrients-17-00276-t003:** In vitro antioxidant activity of the Ohia Lehua honey and Manuka honey.

Samples	DPPH μg TE/g	NO Scavenging Activity mg QE/g	H_2_O_2_ mg TE/g	FRAP μg TE/g
Ohia Lehua honey (0.1 g/mL)	108.33 ± 9.20	85.79 ± 9.45	49.01 ± 4.01	148.85 ± 14.09
Manuka honey (0.1 g/mL)	106.57 ± 13.54	78.83 ± 5.83	44.71 ± 2.79	121.20 ± 11.27

Note: Values are expressed as mean ± SD (n = 3). The honey samples were compared by one-way ANOVA test. DPPH—DPPH free radical scavenging capacity; FRAP—ferric reducing antioxidant power; H_2_O_2_—hydrogen peroxide scavenging capacity; NO—nitric oxide radical scavenging assay; TE—TROLOX equivalent; QE—quercetin equivalent.

**Table 4 nutrients-17-00276-t004:** In vivo antioxidant activity of the Manuka honey and Ohia Lehua honey.

GROUPS	TAC (mmol Trolox Equiv./L)	TOS (µmol H_2_O_2_ Equiv./L)	OSI	NO (µmol/L)	MDA (nmol/L)	AOPP (µmol/L)	SH (µmol/L)
CONTROL	1.11 ± 0.00	27.56 ± 4.35	24.81 ± 3.91	52.94 ± 10.48	4.61 ± 0.46	25.75 ± 2.11	458.2 ± 94.95
INFL	1.10 ± 0.00 ^a^	45.68 ± 6.39 ^aaa^	41.24 ± 5.73 ^aaa^	71.81 ± 16.16	5.80 ± 1.13 ^a^	41.48 ± 3.75 ^aaa^	299.4 ± 53.72 ^aa^
DICLO	1.10 ± 0.00	35.7 ± 3.42 ^b^	32.23 ± 3.06 ^b^	66.41 ± 9.27	5.15 ± 0.35	35.088 ± 4.57 ^b^	356.2 ± 72.07 ^b^
TROLOX	1.11 ± 0.00 ^b^	29.32 ± 5.50 ^bb^	26.40 ± 4.94 ^bb, c^	45.21 ± 4.54 ^bbb, cc^	5.71 ± 0.30	33.25 ± 0.89 ^bb^	412.5 ± 45.35 ^bb, c^
OLH100	1.11 ± 0.00 ^bb; cc^	24.73 ± 4.61 ^bbb, cc^	22.19 ± 4.2 ^bbb, cc^	68.79 ± 11.63 ^dd^	4.48 ± 0.58 ^bb, cc, ddd^	40.26 ± 9.47 ^c, d^	398.6 ± 54.74 ^bb^
OLH50	1.11 ± 0.00 ^bb; cc; d^	26.40 ± 4.21 ^bbb,cc^	22.97 ± 2.68 ^bbb, cc^	69.01 ± 7.8 ^dd^	4.32 ± 0.35 ^bb, cc, ddd^	40.18 ± 7.54 ^c, d^	373.5 ± 50.79 ^bb^
OLH25	1.12 ± 0.00 ^bbb; ccc; ddd^	29.91 ± 5.69 ^bbb^	26.64 ± 5.05 ^bb^	60.30 ± 12.68 ^dd^	3.98 ± 0.43 ^bb, cc, ddd^	41.62 ± 7.91 ^c, d^	369.8 ± 59.20 ^bb^
MH100	1.10 ± 0.00 ^ddd^	45.2 ± 4.54 ^cc, ddd^	45.67 ± 3.30 ^ccc, ddd^	52.49 ± 20.56 ^bb, c^	5.00 ± 0.74	30.11 ± 5.43 ^bb^	462.6 ± 66.84 ^bbb, cc^
MH50	1.10 ± 0.00	41.12 ± 7.13 ^c,d^	37.08 ± 6.44	51.20 ± 19.24 ^bb, c^	5.18 ± 0.22	29.54 ± 2.29 ^bb^	424 ± 58.37 ^bbb, cc^
MH25	1.10 ± 0.00 ^dd^	33.99 ± 7.26 ^bb^	30.72 ± 6.55 ^bb^	54.43 ± 10.22 ^bb,c^	5.14 ± 0.822	32.17 ± 4.94 ^bb^	362.5 ± 60.71 ^bbb,c^

Note: Values are expressed as mean ± SD (n = 5). Vs CONTROL: ^a^ *p* < 0.05, ^aa^ *p* < 0.01, ^aaa^
*p* < 0.001; vs. INFl: ^b^
*p* < 0.05, ^bb^ *p* < 0.01, ^bbb^
*p* < 0.001; vs. DICLO: ^c^ *p* < 0.05, ^cc^ *p* < 0.01, ^ccc^ *p* < 0.001; vs. TROLOX: ^d^ *p* < 0.05, ^dd^ *p* < 0.01, ^ddd^ *p* < 0.001; OLH100—Ohia Lehua honey 100%; OLH50—Ohia Lehua honey 50%; OLH25—Ohia Lehua honey 25%; MH100—Manuka honey 100%; MH50—Manuka honey 50%; MH25—Manuka honey25%; DICLO—Diclofenac; INFL—Inflammation; TAC—Total antioxidant capacity; TOS—Total oxidative status; OSI—Oxidative stress index; NO—Nitric oxide; MDA—Malondialdehyde; AOPP—Advanced oxidation protein products; SH—total thiols.

## Data Availability

Data are available on request.

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
