# Peer review of "Phytochemical Composition and Antioxidant Activity of Manuka Honey and Ohia Lehua Honey"

_nutrients, 2025, doi:10.3390/nu17020276_

Round 1

Reviewer 1 Report

Comments and Suggestions for Authors

Major:

1. Include the standard TAC, TOS, OSI, and phenolic content quantification standard curves in supplementary materials or the main text.

2. Provide a validation section that details how the assay methods were validated (e.g., recovery tests, inter-assay variability)

3. Elaborate on the potential chemical or structural differences between MH and OLH phytochemicals that could influence their respective activities.

Minor:

1. Ensure uniformity in unit representation (e.g., µmol/L, mmol/L) throughout the manuscript.

Comments on the Quality of English Language

Minor linguistic and stylistic corrections are required.

Reviewer 2 Report

Comments and Suggestions for Authors

The article titled: Phytochemical Composition and Antioxidant Activity of Manuka Honey and Ohia Lehua Honey is an interesting paper that introduces information about the antioxidant potential of two types of honey. The results are valuable because they present new data on the antioxidant potential of Ohia Lehua honey. However, before publication, I insist that the following amendments be introduced, which I include below:

·  I suggest extending the keywords to increase the visibility of the article.

· The aim of the work noted at the end of the Introduction could be more detailed. Moreover, the background concerning the biological activity of the examined honeys should be added.

·  It might be interesting to add photos of the samples of the examined honeys (perhaps in supplementary materials) – MH, OLH.

·  Some Latin names should be written in italics (e.g., Pseudomonas aeruginosa line 84, Apis mellifera line 396). Please verify the article for similar mistakes.

·  The abbreviations for Manuka Honey and Ohia Lehua Honey should be repeated in the main text. Once introduced, abbreviations must be used consistently (e.g., in line 456, the abbreviation should be used instead of being introduced again).

· The formula: DPPH IC50 = [(A control - A sample)/A control] × 100 is incorrect. Instead of IC50, it should be % of activity. IC50 can be defined using graphical interpretation of a graph or the curve equation.

· The formula: FRAP IC50 was calculated with the formula: [(A control − A sample)/A control] × 100 is incorrect. The FRAP analysis is conducted without considering the maximum point of absorbance - there are not "100%".

· For FRAP, Hydrogen Peroxide (H2O2) Scavenging Activity, as well as in point 2.5.3. Assessments of Oxidative Stress Markers (excluding MDA analysis), corrections are needed and more detailed methodologie is necessary.

·   The chromatograms are of poor quality. Please replace them with higher-quality versions.

·  I propose standardizing the number of decimal places to two in the results (currently, there are sometimes three and sometimes two).

·  The authors mention high correlation when discussing PCA results. Could the correlation coefficient value be included in the supplementary materials? I also feel that the discussion of OLH results (line 372) and MH results (line 382) is interrupted by the discussion of in vivo antioxidant studies (line 375). Please verify this.

·   In Figure 2, some names overlap, making the figure difficult to read. Please revise.

Reviewer 3 Report

Comments and Suggestions for Authors

The work by Morar et al. concerns Phytochemical Composition and Antioxidant Activity of Manuka Honey and Ohia Lehua Honey. However, some changes have to be entered into the revised version of the manuscript before it can be further processed:

1.     there is no statistical analysis whether the OLH and MH results are statistically different or similar, as it was done e.g. in table 4

2.     There is no description under Table 4 of what statistical test was used for statistical evaluation

3.     there is no PCA analysis between activity and content of active compounds

4.     present innovations in research conducted

5.     present the future perspective of using the presented results. Are there any limitations to the presented research?

Reviewer 4 Report

Comments and Suggestions for Authors

The article "Phytochemical Composition and Antioxidant Activity of Manuka Honey and Ohia Lehua Honey" is focused on evaluating two different honey samples as a potential source of antioxidants and inflammatory-process mediators. To this purpose, the characterization of the phenolic profile of both honeys, as well as a deep evaluation of the antioxidant and anti-inflammatory potential of honey samples through multiple in vitro and in vivo assays have been conducted. The work is generally well-constructed, and interesting for the readers. Therefore, I consider the work suitable to be published in the selected journal, but minor revisions should be made before publication: 

- The authors have been Lines 48-50: Could you provide some quantitative information of honey composition?

- Line 81: Could you specify some phenolics and flavonoids reported in honey?

- Lines 88-89: A deeper review of previous research on these two types oh honeys is missed. 

- Line 185: Put the formula as an equation (same for lines 193, 201, 210). How was the control prepared (same for line 193)?

- Tables 1, 2, and 3: Please include the statistical analysis to know if differences between samples are or not significantly different. 

- Figure 1: Makes numbers bigger to better visualization. 

- Section 3.1.2.: Why concentrations are higher in Manuka honey is signals are greater (Figure 1, according to the axis values) for Ohia honey?

- Line 315: What could be the reason for having found so big differences in phenolic content between both honey samples when analysing by HPLC, but small differences through spectrophotometric assay?

- Table 3: Include in the table the values obtained for the positive controls used in each assay.

- Include in the final section a brief review on limitations of the work and future works. Have the authors considered the possible negative effects caused by others compounds in honeys on the bioactive compounds exerting the described effects? What about bioavailability, and cytotoxic effects of these compounds at the real concentrations exerting the effects?

Round 2

Reviewer 1 Report

Comments and Suggestions for Authors

The authors have thoroughly addressed all my comments.

The only thing I would suggest the authors change is that in Table 2, if the marked presence of a given compound is at the level of 0.00, I recommend entering ND as not detected. The fact that it was not possible to mark a given compound does not mean that it is not present, considering the sensitivity of each analytical method and the determined LOD and LOQ level, indicating the amount of a given compound as 0.00 is a slight distortion.

Reviewer 2 Report

Comments and Suggestions for Authors

I thank the Authors for the amendments they have made.

Reviewer 3 Report

Comments and Suggestions for Authors

1. Table 3 lacks statistical analysis

2. PCA analysis (figure 2) has been added to the manuscript, but the comment is missing
